# Controlling Behavior, Sex Bias and Coaching Success in Japanese Track and Field

**DOI:** 10.3390/sports11020032

**Published:** 2023-01-30

**Authors:** Yuka Tsukahara, Hiroshi Kamada, Suguru Torii, Fumihiro Yamasawa, Aleksandra Katarzyna Macznik

**Affiliations:** 1Department of Sports Medicine, Tokyo Women’s College of Physical Education, Tokyo 1868668, Japan; 2Medical Committee, Japan Association of Athletics Federations (JAAF), Tokyo 1600013, Japan; 3Department of Orthopaedic Surgery, University of Tsukuba, Tsukuba 3058577, Japan; 4Faculty of Sport Sciences, Waseda University, Tokorozawa 3591192, Japan

**Keywords:** overcontrolling, relative energy deficiency in sports, sex bias

## Abstract

Coaching athletes is a complex and lengthy process. Recently, attention has been given to coaches over-controlling behavior toward the athletes’ personal lives and possible sex bias, but the impact of these behaviors on coaching success is unclear. An anonymous survey was answered by 412 track and field coaches (male: 369; female: 43), comprising questions regarding controlling behaviors, sex bias, and personal background. A Chi-square test and logistic regression were performed to determine the factors related to the coach’s characteristics and their success in coaching athletes (to national vs. non-national level). The results showed that controlling behaviors and sex-bias-related beliefs were present. The coaches who coached national-level athletes were more likely to be older, more experienced, and were national level athletes themselves. More national-level coaches reported controlling behaviors but fewer held sex bias beliefs than the non-national level coaches. However, the strength of these beliefs (scores for controlling behavior and sex bias) was not related to the coaching success.

## 1. Introduction

Coaching skills and training methods are not the only factors influencing the outcomes for the athlete. The coach–athlete relationship and the coach’s opinions and held beliefs may affect an athlete’s performance in various ways. In aesthetic and predominantly subjectively judged sports, such as gymnastics, for instance, body size has been associated with the quality of the athlete–coach relationship [1]. While most sports coaching pedagogy includes technical guidance, mentorship, and advice [2], certain coaching behaviors may be excessively controlling and potentially injurious to athletes [3]. If these go too far, coaches may become over-controlling, or even abusive [3,4]. Though excessively restrictive coaching behaviors have been shown to incite conflicts between athletes and coaches and not improve performance [5,6], some coaches still believe these behaviors are vital to successful sports environments [4,7].

Excessively controlling the athlete could also have an effect not only during but also after the athletic career, physiologically and psychologically [4] For example, if the coaches restrict the athletes’ food intake, it could lead to relative energy deficiency in sports (RED-S) and studies have shown that athletes who have suffered from RED-S could be at increased risk of cardiovascular diseases due to altered cholesterol metabolism [8,9]. Moreover, athletes with RED-S, especially during adolescence, are prone to low bone mineral density and decreased peak bone mass which is a predictor for bone fractures [10,11,12]. In addition, studies have shown that, possibly due to excessive pressure from coaches, athletes could experience depressive symptoms [13,14,15,16,17]. Thus, the way coaches coach their athletes, including possible over-controlling, introduces significant health-related consequences to athletes in both the short and long term. Unfortunately, the targeted studies investigating these aspects of the coach–athlete relationship are sparse.

In addition to coaching behaviors, social biases related to age, sex, race/ethnicity, and other individual factors, can influence an athlete’s experience in the sport. As sport is still a male-dominated sector in many aspects [18,19], females face various barriers in the way they are treated and tend to be narrowly defined and valued for their appearance and sexuality [20,21]. These biases and stereotypes can be harmful to athletes’ well-being [22]. Conscious and unconscious sex bias may influence coaching methods and behaviors, which could ultimately have an effect on their health issues. For example, even though the results of both sexes were comparable in skiers, coaches have assumed that men are more competitive than women [23]. Thus, it is speculated that some coaches change their coaching style according to the athlete’s sex, and this may influence the athlete’s performance. However, the prevalence of sex bias in coaches toward their athletes has not been investigated in Japan, although Japan has a very high gender gap in many aspects of society [24].

Thus, the purpose of this study was to investigate the frequency of controlling behaviors and sex biases among Japanese track and field coaches, compare differences between successful and unsuccessful coaches, and identify the factors that are associated with success.

## 2. Materials and Methods

A cross-sectional study of track and field coaches who are associated with the Japan Association of Athletics Federations (JAAF) was conducted. The coaches were recruited via email from JAAF and received an online anonymous survey in January 2021. This study was conducted at the authors’ affiliated institutions and performed according to the guidelines of the Declaration of Helsinki. The project was approved by the Ethics Review Procedures Concerning Research with Human Subjects Group of the authors’ affiliated institution (approval number, 20190170). All participants read and signed the informed consent form before participating.

### 2.1. Participants

All JAAF-authorized coaches who were involved in track and field (*n* = 5241) received a survey via email.

### 2.2. Survey

The survey draft was reviewed by a panel of experts, including the chairs of the coaching committee and medical committee of JAAF. The revised anonymous survey was distributed, and coaches were asked to answer questions regarding their personal background, including sex, affiliation, years of coaching, whether the coaches themselves were track and field athletes, and their level as an athlete. The survey also included questions regarding “controlling” behaviors toward their athletes: whether they restrict romantic relationships (dating), coach male/female athletes regarding weight control and body composition, restrict food intake of male/female athletes, coach male/female athletes to take supplements, and coach athletes to receive iron. We also asked coaches to guess the percentage of coaches who restrict romantic relationships in their athletes and the percentage of coaches who encourage athletes to receive iron injections. Regarding questions assessing sex bias, we asked the coaches whether they change their coaching styles as per the athletes’ sex, whether they think one sex is better than the other at taking care of themselves, whether they think a certain sex is more reliant and less independent, and whether they believe that the sex of the coach is an advantage or disadvantage when coaching athletes of the same or opposite sex. The coaches were also asked whether they coached a national level athlete (or above) to determine their success level in coaching. For the purpose of this study, coaches who coached a national level athlete were considered successful and coaches who coached athlete at a non-national level were considered unsuccessful.

### 2.3. Statistical Analysis

All data were analyzed using the Stata 16.1 (Stata Corporation, College Station, TX, USA). A probability *p*-value < 0.05 was considered significant. Chi-square analysis was performed to find the difference in the proportion of controlling behaviors and sex biases between the national and non-national level coaches. For that purpose, all positive answers were added up and converted to percentages. Multivariate logistic regression was performed to determine the relationship between coaches that coached athletes on a national level and coaches’ background, the score for the strength of controlling behaviors, and the score for the strength of sex bias factors as dependent variables. The scoring of the strength of controlling behaviors and sex bias factors is described in the footnotes of Table 4. We have set male as the base level for sex, and as for the other factors, we have set the ones that had the most responses as the base level except for affiliation and club teams. Odds ratios (OR) and associated 95% confidence intervals were calculated to determine the strength of the model. Cohen’s d was calculated for measuring the effect size.

## 3. Results

A total of 412 coaches participated in the survey (response rate 7.9%), 89.6% males and 10.4% females. Almost all coaches (93.5%) were track and field athletes themselves, and half were a national or international level athlete. A majority (91%) of the coaches coached both male and female athletes in all track and field disciplines, 57.8% coached in schools, and 71.4% coached athletes at a national level. Characteristics of the participants are listed in Table 1.

Although only 7.9% of the coaches admitted that they restrict or try to restrict athletes’ dating themselves, when asked to guess what percentage of coaches restrict dating, most estimated the rate to be 10–50%. Approximately 76% and 79% of the coaches answered that they advise their female and male athletes, respectively, regarding body weight and composition. Moreover, 35.9% and 33.0% of the coaches answered that they restrict the food intake of their female and male athletes, respectively. Approximately half of the coaches answered that they advise their athletes to take supplements, but only 9.1% knew the names and ingredients of these supplements. The majority of coaches reported having some knowledge about anti-doping. Although only 8.2% of coaches answered that they recommend their athletes to receive iron injections, 41.3% of the coaches answered that more than 30% of the coaches recommended their athletes to receive iron injections. The results of the survey are presented in Table 2.

Most coaches (70.9%) admitted that they changed their coaching style based on the athlete’s sex but did not perceive sex (their or the athlete’s) as an advantage or disadvantage. Interestingly, 60% of coaches viewed females as more reliant on others (less independent), but only 6% viewed males in this manner. Most coaches (72.6%) thought that neither males nor females take better care of themselves. Detailed results are presented in Table 2.

Chi-square analysis revealed that significantly more national level coaches than non-national coaches reported coaching both male and female athletes on their body weight and composition, restricting the food intake of female athletes, and coaching male athletes to take nutritional supplements. Compared to national level coaches, significantly more non-national level coaches answered that a certain sex of the athlete is better than the other at taking care of themselves and that coaches’ sex can be an advantage or disadvantage while coaching female athletes, but not male athletes. There were no differences in the number of coaches coaching athletes to take iron injections. The results of the chi-square analysis are presented in Table 3.

The results of the logistic regression analysis revealed that the coach’s sex and affiliation were unrelated to the athlete’s performance level (national vs. non-national), but coaches’ experience was negatively related to the coaches’ level, meaning that coaches with limited experience were not able to coach athletes at a national level compared to coaches with more experience (*p* < 0.001). In addition, coaches’ level as an athlete (of at least national level) was positively related to their athlete’s level (OR 2.49, *p* = 0.002). The scores for the strength of controlling behaviors and sex bias were not related to the coach’s level. The results of the logistic regression analysis are demonstrated in Table 4.

## 4. Discussion

This is the first study to investigate the frequency of coaches’ controlling behaviors and sex bias and the possible association of various factors affecting coaching success in Japanese track and field. The results show that controlling behaviors and sex-bias-related beliefs are present in the coaches. Moreover, the proportion of these behaviors and beliefs differs between national level (successful) coaches and non-national level (unsuccessful) coaches, suggesting that they may have an influence on coaching success. Coaches’ experience as a national level athlete (or above) and 10 years of coaching experience were associated with coaching success. The results of this study provide useful insides into what makes the coach successful or not.

### 4.1. Controlling Behaviors

Controlling behaviors are present in Japanese track and field coaches. In fact, they are more common in successful coaches (coaches who coached athletes to the national or international level). The score of the strength of controlling behaviors (weighted sum of the answers to questions on controlling behaviors) was not, however, associated with coaching success.

Although a few coaches in this study admitted that they restrict dating and romantic relationships in their athletes, most thought that this behavior is present in a high percentage of coaches. The relationship between sports performance and the romantic status of an athlete has been investigated in previous studies, and Campbell et al. have reported that Olympic athletes have answered that they performed better when in love [25]. On the other hand, being in a romantic relationship during their athletic career was also considered to have negative effects on sports performances due to conflicts, jealousy, and mood swings [26]. In Japan, it is widely believed that “if you are in love, you will not perform well,” and “you cannot juggle the athlete’s role and being in a romantic relationship” [26]. This aspect of the coach–athlete relationship certainly requires further investigation.

Most coaches coached their athletes, both male and female, regarding their body weight and composition, and significantly more national coaches engaged in this behavior compared to non-national coaches. Moreover, one-third of coaches restricted the food intake of their athletes, with relatively more national level coaches engaging in this behavior but only with their female athletes. The possible disadvantages of restricting the food intake of athletes and limiting energy availability leading to RED-S have been established in previous studies [27,28,29]. Low energy availability negatively affects performance and leads to oligomenorrhea/amenorrhea in female athletes, which can cause low bone mineral density [9,30]. Low energy availability in male runners was reported to cause bone stress injuries [31]. Additionally, it has been reported that there were no correlations between body weight and performance in track and field athletes [32]. Therefore, limiting food intake may have serious negative consequences that go beyond athletic performance.

On the other hand, coaching athletes regarding body weight and composition may not be focused on restricting food intake but rather on properly nourishing athletes’ bodies. More national level coaches admitted to this behavior, which may indicate that some coaches’ advice and guidance on the topic of body composition and body weight may contribute to coaching success.

The current study found that half of the coaches advised their athletes to take supplements, but only 9.1% reported that they know the details of the supplements their athletes are taking. The advantages of nutritional supplements on the performance of the athlete with no nutritional deficiency are still questionable [33,34,35]. Less than 30% of coaches believed that they have good knowledge of anti-doping. Moreover, some nutritional supplements pose a risk of being contaminated with non-approved ingredients which could cause harm to the athlete’s health or cause anti-doping rule violations [36,37]. In this study, the prevalence of recommending supplements was significantly higher among national level coaches than the non-national level coaches regarding male athletes. A similar observation was also reflected in the data presented by Knapik et al., wherein the higher use of supplements was seen in the elite athletes than the non-elite counterparts [35]. As coaches’ self-reported knowledge of supplements is low, the advice on this topic should be questioned if not restricted.

Similar concerns apply to iron injections. Although 92% of the coaches answered that they have never recommended their athletes to receive iron injections, they also estimated that approximately 80% of coaches are recommending the injections. Interestingly, it has been previously noted that Japanese long-distance runners did not recognize the risks of injections or did not mind coaches recommending the injections [38]. Risks of receiving inappropriate iron injections include liver failure [38,39,40]. If almost 80% of coaches recommend iron injections (without proper medical knowledge), it puts the athletes’ health at risk and should be addressed immediately. The results of this survey showed that education about the risks related to supplements and iron injections should include both coaches and athletes to bring about the required change.

In summary, this study has shown that some controlling behaviors are not inherently wrong. Indeed, they may take the form of education and guidance, and can be helpful to the athlete and possibly enhance their performance. On the other hand, advice on interventions that pose a high risk, e.g., unnecessary iron injections, should be delegated to medical professionals

### 4.2. Sex Bias

Beliefs related to sex bias are present in Japanese track and field coaches. They are, however, significantly less common in national coaches than in non-national coaches. Most of the coaches (73%) thought that neither female nor male athletes take better care of themselves, and most coaches did not think that the coach’s gender matters. This is in line with the previous study showing that track and field high school athletes did not display sex bias towards the coaches [41]. However, another study has found that there are actually significant differences between female and male athletic coaches [42].

However, in this study, 70% of the coaches have reported that they adopt their coaching practice to the athlete’s sex, and 60% of the coaches believed that female athletes are more reliant on others.

Although sex bias certainly is present, it is less prevalent at the national level; and some aspects of it, e.g., adjusting coaching strategies, may be helpful in enhancing athletic performance levels.

### 4.3. Other Differences between Successful and Unsuccessful Coaches

We have found a number of factors relating to coaching success, as measured by their athletes achieving the national (or above) level of athletic performance (success). The factors that were associated with coaching success were the coach’s age, years of coaching, and the level the coach has reached during their own athletic career. Our results indicated that coaches with less experience, especially those below 10 years, were more likely to be unsuccessful. Coaches who used to be an athlete competing at a national level (or higher) were 2.5 times more likely to be coaching a national level athlete.

Coaches’ support has been shown as vital for athletes to develop higher self-esteem [43,44]. Thus, in this study, it is speculated that it takes a considerable amount of time not only to acquire the skills, but also to build a successful relationship between an athlete and a coach. In addition, coaches’ level as a former athlete positively correlated with their athlete performance level. This could have been related to the coach having experience and knowledge on what exactly is required to perform at the national level and beyond. Other aspects could be related to motivation, encouragement, and leading by the example of these coaches, allowing the athletes to look up and follow the coach’s guidance rather than be controlled or forced.

Additionally, when the controlling behaviors and sex-bias-related beliefs were scored according to their strength and number present in a coach, they were found to not influence the athletes’ performance level. This may mean that the fact that the coach has any sex bias is more important than how many of these beliefs the coach holds. Moreover, as discussed above, some of the behaviors and beliefs, even though commonly named as controlling or biased, may be beneficial to athletes when grounded in their best interest or based on science and experience.

This study has several limitations. First, although it was an anonymous survey, it relied on self-reporting, and therefore it is possible that the respondents did not answer the questions truthfully. Second, there was a selection bias since the response rate was low and the survey responders were 90% males, working mostly with junior high and high school athletes. Therefore, the results may not be directly generalizable to other populations of coaches and/or athletes. Third, controlling behaviors and sex bias in coaches were investigated using a self-report on a custom-prepared survey in this study. Therefore, further investigations using validated tools and a broader range of perspectives (including athletes’ perspective) are required. Last, the frequency of particular behaviors differed drastically depending on whether the coaches reported on themselves versus other coaches. This warrants further investigation.

## 5. Conclusions

This study indicates that successful Japanese track and field coaches (who coached at least national level athletes) reported more controlling behaviors but fewer sex-bias-related beliefs when compared to less successful coaches (who coached athletes at below national level). Coach’s experience of more than 10 years and previous athletic career at national level or higher were identified as factors related to their success. The present study sheds light on the possible influence of controlling behaviors and sex bias on coaches’ success. A deeper understanding of these areas will provide the basis for the development of effective strategies for safer and more successful participation in sports for both coaches and athletes.

## Figures and Tables

**Table 1 sports-11-00032-t001:** Characteristics of the coaches.

	%	*n*
Gender		
Male	89.6	369
Female	10.4	43
Affiliation		
Elementary school	6.1	25
Junior high school	15.5	64
High school	31.3	129
University/College	4.9	20
Club teams	25.5	105
Corporate team	3.6	15
Others	13.1	54
Age		
≤19 years	2.7	11
20–29 years	4.9	20
30–39 years	20.6	85
40–49 years	30.3	125
50–59 years	28.2	116
60–69 years	10.7	44
70–79 years	2.7	11
Years of coaching		
≤5 years	11.9	49
5–9 years	18.5	76
10–14 years	18.2	75
15–19 years	12.6	52
20–29 years	20.2	83
30–39 years	14.6	60
≥40 years	4.1	17
Events coached		
Sprinting events	5.3	24
Jumping events	1.6	7
Throwing events	2.2	10
Distance events	12.8	58
All of the above	78.1	353
Coached national level athlete (yes)	71.4	294
Former athlete (yes)	93.5	385
Events as an athlete		
Sprinting events	37.6	143
Jumping events	17.1	65
Throwing events	8.2	31
Distance events	32.1	122
Combined events	5.0	19
Level as an athlete		
International	8.1	31
National	47.4	182
Regional	19.0	73
Prefectural	22.9	88
Hobby	2.6	10
Sex of the coached athlete		
Female only	6.0	24
Male only	3.0	12
Both	91.0	366

**Table 2 sports-11-00032-t002:** Results of the survey.

Question & Answer Options	(%)	*n*
Do you restrict your athletes dating?		
Yes, I restrict them	1.6	6
I try to restrict them	6.3	24
No	92.1	352
What do you think is the % of coaches who restrict their athletes dating?		
Over 50%	14.6	60
30–50%	28.2	116
10–30%	41.0	165
Less than 10%	17.7	71
Do you coach your female athletes regarding body weight and composition?		
Very much	6.6	27
Quite a lot	24.5	101
A little	44.9	185
Not at all	24.0	99
Do you coach your male athletes regarding body weight and composition?		
Very much	7.5	31
Quite a lot	33.7	139
A little	38.1	157
Not at all	20.6	85
Do you tell your female athletes to limit their food intake?		
Very much	0.7	3
Quite a lot	7.8	32
A little	27.4	113
Not at all	64.1	264
Do you tell your male athletes to limit their food intake?		
Very much	0.7	4
Quite a lot	7.5	31
A little	24.5	101
Not at all	67.0	276
Do you tell your female athletes to take supplements?		
Very much	2.2	9
Quite a lot	10.9	45
A little	36.4	150
Not at all	50.5	208
Do you tell your male athletes to take supplements?		
Very much	4.4	18
Quite a lot	15.1	62
A little	35.2	145
Not at all	45.4	187
How well do you know about the supplements the athletes are taking?		
Everything including the names and the ingredients	9.1	37
Know all the names of the product	8.6	35
Know a little of the names and the ingredients	44.0	178
Not at all	38.3	155
Do you have enough knowledge regarding anti-doping?		
Very much	29.1	119
Quite a lot	43.0	176
A little	25.7	105
Not at all	2.2	9
Do you recommend your athletes to receive iron injections?		
Very much	0.2	1
Quite a lot	1.2	5
A little	6.8	28
Not at all	91.8	378
What do you think is the % of coaches who coach their athletes to receive iron injections?		
Over 50%	13.4	55
30–50%	27.9	115
10–30%	37.1	153
Less than 10%	21.6	89
Do you change coaching style by athletes’ sex? (yes)	70.9	292
Who can take care of themselves better, male or female athletes?		
Female athletes	8.7	35
Male athletes	18.7	75
Neither	72.6	292
Who do you think rely on others (not independent), male or female athletes?		
Female athletes	60.0	241
Male athletes	6.0	24
Neither	34.1	137
Female coaches are at an advantage of coaching female athletes compared to male counterparts		
Female coaches are at an advantage	38.9	160
Female coaches are at a disadvantage	2.4	10
Neither	58.6	241
Male coaches are at an advantage of coaching female athletes compared to female counterparts		
Male coaches are at an advantage	6.1	25
Male coaches are at a disadvantage	24.6	101
Neither	69.3	285
Female coaches are at an advantage of coaching male athletes compared to male counterparts		
Female coaches are at an advantage	5.0	20
Female coaches are at a disadvantage	14.2	57
Neither	80.8	324
Male coaches are at an advantage of coaching male athletes compared to female counterparts		
Male coaches are at an advantage	23.2	93
Male coaches are at a disadvantage	0.8	3
Neither	76.1	305

**Table 3 sports-11-00032-t003:** Results of the chi-square analysis; differences between the national level and non-national level coaches.

	National Level (+) (%) *	Non-National Level (−)(+) (%)	*p* Value	Effect Size
Restrict their athletes to have a romantic relationship	8.6	4.5	0.163	0.161
Coach female athletes regarding body weight and composition	81.3	62.7	**<0.001**	0.44
Coach male athletes regarding body weight and composition	82.3	72.0	**0.020**	0.26
Restrict food intake of female athletes	39.8	26.3	**0.010**	0.28
Restrict food intake of male athletes	35.4	27.1	0.107	0.18
Coach female athletes to take supplements	52.0	43.2	0.105	0.18
Coach male athletes to take supplements	58.8	44.1	**0.006**	0.30
Coach athletes to receive iron injections	8.8	6.8	0.491	0.07
Female/male athletes rely on others compared to the other sex athletes	66.0	65.8	0.968	0.06
Female/male athletes are better at taking care of themselves compared to the other sex athletes	24.4	35.1	**0.031**	0.27
Female coaches have advantage or disadvantage when coaching female athletes	37.8	50.4	**0.019**	0.26
Female coaches have advantage or disadvantage when coaching male athletes	18.2	21.8	0.414	0.09
Male coaches have advantage or disadvantage when coaching female athletes	26.5	41.0	**0.004**	0.32
Male coaches have advantage or disadvantage when coaching male athletes	22.0	29.1	0.137	0.17

Bold font indicates statistical significance; * the % depict all the coaches who answered positively to the posed question (sum of all the answers other than ‘no’ or ‘not at all’).

**Table 4 sports-11-00032-t004:** Results of logistic regression analysis; factors that affect whether the coaches were able to coach athletes to a national level.

	Odds Ratio	95% Confidence Interval	*p* Value
Gender			
Male	1.0		
Female	0.79	[0.34, 1.82]	0.589
Age			
40–49 years	1.0		
≤19 years	0.36	[0.07, 1.82]	0.217
20–29 years	0.96	[0.30, 3.09]	0.947
30–39 years	0.76	[0.34, 1.71]	0.508
50–59 years	0.38	[0.17, 0.87]	**0.023**
60–69 years	0.32	[0.11, 0.98]	**0.046**
70–79 years	0.56	[0.07, 4.37]	0.583
Affiliation			
Club team	1.0		
Elementary school	0.67	[0.21, 2.14]	0.503
Junior high school	2.15	[0.84, 5.53]	0.113
High school	0.66	[0.30, 1.46]	0.303
University/College	3.04	[0.35, 26.26]	0.313
Corporate team	4.38	[0.51, 37.78]	0.179
Others	0.75	[0.32, 1.71]	0.488
Coaching years			
20–29 years	1.0		
≤5 years	0.11	[0.04, 0.35]	**<0.001**
5–9 years	0.11	[0.04, 0.30]	**<0.001**
10–14 years	0.48	[0.17, 1.38]	0.174
15–19 years	0.36	[0.13, 1.05]	0.062
30–39 years	1.22	[0.40, 3.80]	0.726
≥40 years	0.94	[0.14, 6.16]	0.951
Level as an athlete			
Non-national level	1.0		
At least national level	2.49	[1.39, 4.48]	**0.002**
Score for controlling factors *	0.96	[0.88, 1.04]	0.324
Score for sex bias factors *	0.89	[0.75, 1.05]	0.172

Bold font indicates statistical significance; * Score for controlling factors was a sum of the scores for questions asking about controlling behaviors (Do you restrict your athletes’ dating? Yes, I restrict them = 2 points, I try to restrict them = 1 point, No or others = 0 points; Do you coach your female athletes regarding body weight and composition? Very much = 3 points, quite a lot = 2 points, a little = 1 point, and not at all = 0 points; Do you coach your male athletes regarding body weight and composition? Very much = 3 points, quite a lot = 2 points, a little = 1 point, and not at all = 0 points; Do you tell your female athletes to limit their food intake? Very much = 3 points, quite a lot = 2 points, a little = 1 point, and not at all = 0 points; Do you tell your male athletes to limit their food intake? Very much = 3 points, quite a lot = 2 points, a little = 1 point, and not at all = 0 points; Do you tell your female athletes to take supplements? Very much = 3 points, quite a lot = 2 points, a little = 1 point, and not at all = 0 points; Do you tell your male athletes to take supplements? Very much = 3 points, quite a lot = 2 points, a little = 1 point, and not at all = 0 points; Do you recommend your athletes to receive iron injections? Very much = 3 points, quite a lot = 2 points, a little = 1 point, and not at all = 0 points); Score for gender bias factors was a sum of scores for the questions regarding sex bias (Do you change coaching style by athletes’ sex? yes = 1 point, no = 0 points; Who can take care of themselves better, male or female athletes? Female/male = 1 point, neither = 0 points; Who do you think relies on others (not independent), male or female athletes? Female/male = 1 point, neither = 0 points; Female coaches are at an advantage of coaching female athletes compared to male counterparts? Advantage/disadvantage = 1 point, neither = 0 points; Male coaches are at an advantage of coaching female athletes compared to male counterparts? Advantage/disadvantage = 1 point, neither = 0 points; Female coaches are at an advantage of coaching male athletes compared to female counterparts? Advantage/disadvantage = 1 point, neither = 0 points; Male coaches are at an advantage of coaching male athletes compared to female counterparts? Advantage/disadvantage = 1 point, neither = 0 points).

## Data Availability

Data are available upon reasonable request from the corresponding author via email.

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
