# Peer review of "Controlling Behavior, Sex Bias and Coaching Success in Japanese Track and Field"

_sports, 2023, doi:10.3390/sports11020032_

Round 1

Reviewer 1 Report

In summary, this is a cross-sectional survey study investigating how the coach's controlling behavior and sex bias may influence athlete's performance. While the intention of the study is good, the study itself suffers from low response rate, selection bias, and extrapolation. Also, lengthy discussion of irrelevant findings are distracting. The authors would need to acknowledge limitations of this study more. Please see below for further comments.  

Title: result on athlete's health is barely mentioned. 

Line 18-20: The result here is somewhat irrelevant with the aim of this study. After reading the whole manuscript, it seems that this result was found from the multiple logistic regression, and since it was not the primary goal, this result can be omitted from the abstract.

Introduction is overall well written.

Line 78-79: Please explain how the survey was validated. Please attach the sample of study as a supplemental file.

Line 96: Can you clarify the outcome (dependent variable) for the multiple logistic regression and justify why it was performed? 

Line 165: was coach's level as a former athlete also part of primary aim? Based on the introduction, the primary aim was to investigate the effect of controlling behaviors and sex bias on athletes' performance. 

Line 169-170: but the result doesn't support that those behaviors affect performance. 

Line 170-172: These statements rather belong to conclusion or later in the discussion when mentioning future research direction

Line 165-172: Briefly summarize the findings of the current study rather than making vague statements that are not supported by the current results.

Line 251-270: This was not the primary aim of this study. I think discussion should stick to controlling behavior and sex bias unless the authors' have clear justification for discussing all these factors. This part is rather distracting to overall manuscript. 

Line 297: There are more limitations: very low response rate, potential selection bias - there were more coaches that coached national level. I assume overall there are more coaches that coached non-national athletes. Also, the actual result states that only 7.5% of the coaches admitted that they restrict athletes' dating...(line 115-116) and 8.2% recommended athletes to receive iron....(line 122-124), but all of sudden they THINK that other higher coaches are doing something else. This is also bias. Authors should admit that they are extrapolating based on what the coaches THINK others are doing rather than based on actual survey results. Furthermore, there may be more controlling behaviors or sex bias that the survey failed to ask. 

Line 307: have a subheading for conclusion. 

Author Response

Reviewers’ comments are in written in black, authors’ response is written in red.

Authors would like to thank the reviewers for their valuable comments and suggestions.

In summary, this is a cross-sectional survey study investigating how the coach's controlling behavior and sex bias may influence athlete's performance. While the intention of the study is good, the study itself suffers from low response rate, selection bias, and extrapolation. Also, lengthy discussion of irrelevant findings are distracting. The authors would need to acknowledge limitations of this study more. Please see below for further comments.  

RESPONSE:

First of all. Thank you very much for taking your time out of your busy schedule. I have revised the paper based on your recommendations. Again, thank you very much and I really appreciate it.

Discussion has been shortened and limitations’ section extended.

Title: result on athlete's health is barely mentioned. 

RESPONSE

Thank you for your comment.

FROM

How coach’s controlling behavior and sex bias influence athlete’s health and performance in track and field athletes

TO

How Japanese coach’s controlling behaviors and sex bias influence performance level in track and field athletes

Line 18-20: The result here is somewhat irrelevant with the aim of this study. After reading the whole manuscript, it seems that this result was found from the multiple logistic regression, and since it was not the primary goal, this result can be omitted from the abstract.

RESPONSE

Thank you for your comment and we have rewritten the results section of the abstract.

Introduction is overall well written.

RESPONSE

Thank you

Line 78-79: Please explain how the survey was validated. Please attach the sample of study as a supplemental file.

RESPONSE

Thank you for your comment but it was discussed by the panel of experts and was distributed to the coaching committee of JAAF and they have edited the survey after they have tested the survey themselves. I took out the word “validated” to avoid confusion. Also, the details of the questions asked for this paper are all written in Table 1 and 2 (I have edited the tables after some advice from the reviewers)

Line 96: Can you clarify the outcome (dependent variable) for the multiple logistic regression and justify why it was performed? 

RESPONSE

Thank you for your comment and gender, age, affiliation, coaching years, level as an athlete, controlling factors, and gender bias factors were the dependent variables. We wanted to emphasize that although coaches have gender bias and overcontrol their athletes, doing so is not helping to enhance the athletes’ performance and could affect their health. I hope this makes sense.

Line 165: was coach's level as a former athlete also part of primary aim? Based on the introduction, the primary aim was to investigate the effect of controlling behaviors and sex bias on athletes' performance. 

RESPONSE

Thank you for your comment. The primary aim is to investigate the effect of controlling behaviors and sex bias on athletes' performance level.Coach’s level as a former athlete was one of the factors related to the athletic performance level.

Line 169-170: but the result doesn't support that those behaviors affect performance. 

RESPONSE

Thank you for your comment.  We have rewritten this section for clarity.

Line 170-172: These statements rather belong to conclusion or later in the discussion when mentioning future research direction

RESPONSE

Thank you for your comment, these statements have been removed.

Line 165-172: Briefly summarize the findings of the current study rather than making vague statements that are not supported by the current results.

RESPONSE

Thank you for your comment, we have rewritten this section.

Line 251-270: This was not the primary aim of this study. I think discussion should stick to controlling behavior and sex bias unless the authors' have clear justification for discussing all these factors. This part is rather distracting to overall manuscript. 

RESPONSE

Thank you for your comment.

This section was cut down but is of a great importance for the primary aim as shows that the strength of controlling behaviors and sex bias did not relate to performance level in the logistic regression analysis.

The title of the section was changed to: Other differences between successful and unsuccessful coaches

Line 297: There are more limitations: very low response rate, potential selection bias - there were more coaches that coached national level. I assume overall there are more coaches that coached non-national athletes. Also, the actual result states that only 7.5% of the coaches admitted that they restrict athletes' dating...(line 115-116) and 8.2% recommended athletes to receive iron....(line 122-124), but all of sudden they THINK that other higher coaches are doing something else. This is also bias. Authors should admit that they are extrapolating based on what the coaches THINK others are doing rather than based on actual survey results. Furthermore, there may be more controlling behaviors or sex bias that the survey failed to ask. 

RESPONSE

Thank you for your comment and we have made the changes accordingly.

FROM

This study has several limitations. First, although it was an anonymous survey, it is possible that the respondents did not answers the questions truthfully. Second, the survey responders were 90% males, working mostly with junior high and high school athletes, both males and females, and therefore the results may not be directly generalizable to other populations of coaches and/or athletes. Third, while controlling behaviors and sex bias in coaches were investigated using a self-report on a custom prepared questions in this study, further investigations using validated tools and broader range of perspectives (including athletes’ perspective) are required. For example, the Controlling Coach Behaviors Scale or similar tool should be validated in Japanese population and used to provide more robust data [30].

TO

This study has several limitations. First, although it was an anonymous survey, it is possible that the respondents did not answers the questions truthfully. Second, there was a selection bias since the response rate was low and the survey responders were 90% males, working mostly with junior high and high school athletes, and therefore the results may not be directly generalizable to other populations of coaches and/or athletes. Third, while controlling behaviors and sex bias in coaches were investigated using a self-report on a custom prepared questions in this study, further investigations using validated tools and broader range of perspectives (including athletes’ perspective) are required. For example, the Controlling Coach Behaviors Scale or similar tool should be validated in other language including Japanese and used to provide more robust data and more questions related to controlling behaviors and sex bias should have been asked [30]. Last, it is possible that we are extrapolating the results, as only 7.5% of the coaches admitted that they restrict athletes' dating and 8.2% recommended athletes receive iron injections, which are much lower than the responses to "What do you think is the percentage of coaches who restrict athletes' dating/recommend athletes receive iron injection?".

Line 307: have a subheading for conclusion. 

Thank you for your comment and I have added Conclusion subheading.

Reviewer 2 Report

Even though the study is well presented, it is not answering the research question. It is the study of the attitude and behaviour of the coaches. It does not measure athletes' health and performance. So there is no scope for finding such a relationship in this study. The title, aim, analysis, results, and discussion has to be rewritten accordingly.

This study is limited to Japanese track and field coaches only. It cannot be generalized. The title has to be changed accordingly. The title should contain the term 'Japanese,

Author Response

Reviewers’ comments are in written in black, authors’ response is written in red.

Authors would like to thank the reviewers for their valuable comments and suggestions.

Even though the study is well presented, it is not answering the research question. It is the study of the attitude and behaviour of the coaches. It does not measure athletes' health and performance. So there is no scope for finding such a relationship in this study. The title, aim, analysis, results, and discussion has to be rewritten accordingly.

RESPONSE:

First of all. Thank you very much for taking your time out of your busy schedule. I have revised the paper based on your recommendations. Again, thank you very much and I really appreciate it.

We have removed all the suggestions on the influence on athletes’ health apart from the discussion where we speculate risks for restricting of food based the previous research. The controlling behaviours and sex bias as analysed and discussed in the light of statistical methods used, and athlete performance level is broadly divided into national versus non-national level.  

This study is limited to Japanese track and field coaches only. It cannot be generalized. The title has to be changed accordingly. The title should contain the term 'Japanese,

RESPONSE:

The term ‘Japanese’ has been added to the title.

Round 2

Reviewer 1 Report

Authors' response was reasonable, and I appreciate their effort to improve the manuscript.

Author Response

Thank you very much for your time.

Reviewer 2 Report

Even though the authors tried to improve the manuscript, the major issue is not solved and is not addressed properly. This study is a survey study about the attitude and behaviour of the coaches. It is not answering your research question and the aim of the study. How did you measure the controlling behaviour of coaches? How did you measure the performance? If these things are not measured, it is impossible to determine the influence of this behaviour on performance. Only an experimental study can answer this type of research question. I recommend changing the research question and aim and making it a survey on the attitude and behaviour of coaches.

Author Response

Thank you for your comment. The research question/aim has been changed to reflect the design of the study. We have also changed the manuscript’s title and some parts of the manuscript to reflect the change in the stated research question/aim. We have added references to the SEX BIAS section of the discussion as per the reviewer’s request. The major changes of the manuscript are listed below, and in addition to that, the smaller details are listed in the document (track changes). Again, thank you very much for taking your time out of your busy schedule to make this manuscript better.

Title

FROM

How Controlling behavior, sex bias and coaching success in Japanese track and field

TO

Controlling behavior, sex bias and coaching success in Japanese track and field

Introduction (line 127-129)

FROM

Thus, the purpose of this study was to investigate controlling behaviors and sex biases among Japanese track and field coaches, and to understand the relationship between these factors and their athletes’ performance level.

TO

Thus, the purpose of this study was to investigate the frequency of controlling behaviors and sex biases among Japanese track and field coaches, compare differences between successful and unsuccessful coaches and identify the factors that are associated with success.

Methods (survey, line 160,206-209)

FROM

The coaches were also asked whether they coached a national level athlete or not.

TO

The coaches were also asked whether they coached a national level athlete (or above) to determine their success level in coaching. For the purpose of this study, coaches who coached a national level athlete, were considered successful, and coaches how coached athlete at a non-national level were considered unsuccessful.

Discussion (line 605-607)

ADDED

More national level coaches admitted to this behavior, which may indicate that some coaches’ advice and guidance on the topic of body composition and weight may contribute to coaching success.

Discussion (line 605-607)

FROM

Believes related to sex bias are present in Japanese track and field coaches. They are however significantly less common in national coaches when compared with non-national coaches. Seventy percent of the coaches have reported that they adopt their coaching practice to the athlete’s sex. This could have been related to the believe held by 60% of the coaches that female athletes are more reliant on others than male athletes, or other psychological factors researched in other studies [9,41]. On the other hand, most of the coaches (73%) thought that neither female or male athletes take better care of themselves, and most coaches did not think that coach’s gender matters. Therefore, although sex bias certainly is present, it is less prevalent at national level, and some aspects of it like adjusting coaching strategies may actually be helpful in enhancing athletic performance levels.

TO

Beliefs related to sex bias are present in Japanese track and field coaches. They are however significantly less common in national coaches than in non-national coaches. Most of the coaches (73%) thought that neither female nor male athletes take better care of themselves, and most coaches did not think that the coach’s gender matters. This is in line with the previous study showing that track and field high school athletes did not display sex bias towards the coaches [41]. However, another study has found that there actually are significant differences between female and male athletic coaches [42].

However, in this study, seventy percent of the coaches have reported that they adopt their coaching practice to the athlete’s sex, and 60% of the coaches believed that female athletes are more reliant on others.

Although sex bias certainly is present, it is less prevalent at national level; and some aspects of it like adjusting coaching strategies may be helpful in enhancing athletic performance levels.

ADDED (line 1025-28)

Also, as discussed above, so of the behaviors and believes even though commonly named as controlling or biased may be beneficial to athletes when grounded in their best interest or based on science and experience.